# Robust Video Perception by Seeing Motion

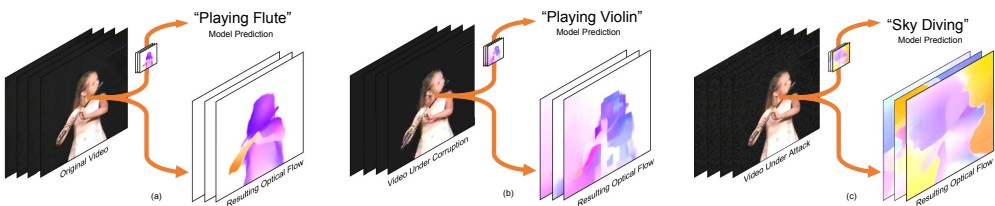

Figure 1: Natural corruptions and adversarial attacks on video perception models not only cause video classifiers to fail, but also collaterally corrupt the motion estimation, captured by optical flow. We show the motion estimation of a normal video in (a), a video corrupted by JPEG compression in (b), and an adverarially attacked video in (c). Our work aims to exploit the inconsistencies between the optical flow and the video to robustify video perception.

## Abstract

Despite their excellent performance, state-of-the-art computer vision models often fail when they encounter shifted distributions or adversarial examples. We find existing video perception models fail because they are not able to perceive the correct motion. Inspired by the extensive evidence that motion is a key factor in the human visual system, we propose to correct what the model sees by restoring the perceived motion information. We create a test-time constraint using motion information without any human annotation, where this constraint should be respected by all robust video perception models. Our key observation is that this constraint is violated when the inputs are corrupted or adversarially attacked. By optimizing the input to respect the constraint at test time, we can adapt the inference to be robust. Visualizations and empirical experiments on UCF-101 and HMDB-51 datasets show that restoring motion information in deep vision models improves robustness under both common noise corruptions and worst-case perturbations.

## 1 Introduction

Deep models have consistently achieved high performance over a large number of computer vision tasks Krizhevsky et al. (2012); Simonyan & Zisserman (2014); He et al. (2017); Ji et al. (2012); Taylor et al. (2010). However, it has been shown that they are susceptible to natural corruptions Hendrycks & Dietterich (2019) and adversarial examplesCarlini & Wagner (2017); Goodfellow et al. (2015); Szegedy et al. (2014), where additive perturbations are optimized to fool the model. This vulnerability introduces problems to real-world scenarios where safety and robustness are crucial Kurakin et al. (2018); Chernikova et al. (2019); Dong et al. (2019). To address this problem, a large body of work has studied robust models for image recognition Madry et al. (2018); Goodfellow et al. (2015); Mao et al. (2019); Zhang et al. (2019a); Cohen et al. (2019); Mao et al. (2021b); Nie et al. (2022); Wu et al. (2021); Mao et al. (2021c; 2022a; 2021a). While robustness on static images achieved significant progress, robust models on the more realistic video data are under-explored due to the complexity Xiao et al. (2019).

Learning robust video perception models is challenging because the input dimensionality of video models is high, which provides the attacks and corruptions more space to manipulate in the input space Simon-Gabriel et al. (2019). Data augmentation Goodfellow et al. (2015); Madry et al. (2018); Mao et al. (2021a) has been studied to improve video models' robustness Kinfu & Vidal (2022). However, they need to anticipate the type of corruption and attacks to train on them, which causes these models to be vulnerable to unforeseen attacks Zhang et al. (2019b).

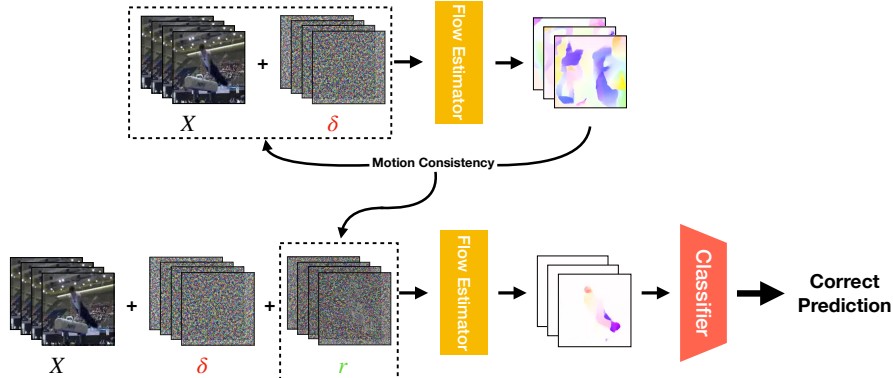

Figure 2: **Our method.** Common and adversarial perturbations ($\delta$) on RGB inputs result in inaccurately estimated motion, indicated by the distorted optical flow. Distorted optical flow provides a self-supervised consistency loss measured by the warping error. Optimizing for this consistency allows us to obtain a reverse perturbation $r$ that restores the temporal consistency and corrects misclassified results.

In this paper, we introduce an approach for robust video perception by adding an additional temporal constraint, which has been ignored by most video perception algorithms. Despite having high input dimensionality, video data is highly structured along the temporal dimension. While natural corruptions and adversarial attacks damage the video classifier, we find that they also simultaneously destroy the temporal structure in the time dimension. We will show how to leverage this intrinsic structure to create robust video classifiers, where we repair this temporal constraint at inference time.

One important temporal structure in videos is motion, which is captured by the optical flow Ng et al. (2018); Sevilla-Lara et al. (2018); Inkawhich et al. (2018). We find that the optical flow under many natural corruptions and attacks is damaged. We create a self-supervised task using this motion, where we warp the current video frame with the flow field to construct the future video frame. We then use the prediction error of the future video frame as our test-time constraint that the video model should respect.

We propose to restore the motion-based constraints at test-time, which enables us to dynamically adapt the model to unforeseen distribution shifts and corruptions. Our method does not require training the video model on a number of possible corruptions and attacks, which lifts the burden of training a video classifier.

Visualizations and empirical results on UCF-101 and HMDB-51 datasets show that enforcing motion consistency improves models' robustness under different natural noise corruptions by up to 6.5 points, and adversarial attacks by up to 72 points. Since our method is at inference time, it is compatible with existing training-based robust models, and the user can choose whether to perform this defense based on need. We will release our model, data, and code.

## 2 RELATED WORK

**Motion Estimation**. Optical flow is the standard way to capture motion. Classical variational methods Horn & Schunck (1981); Lucas et al. (1981); Mémin & Pérez (1998); Brox et al. (2004); Zach et al. (2007) for estimating optical flow are based on the minimization of an energy function, which typically consists of a photometric consistency term and a smoothness term. Recently, convolutional neural network-based methods demonstrate better optical flow estimation Dosovitskiy et al. (2015); Ilg et al. (2017); Sun et al. (2018); Teed & Deng (2020). These approaches usually directly regress the ground truth optical flow calculated from synthetic data in a supervised manner. More closely related to our work, deep unsupervised optical flow estimation has also been an interesting line of research Jonschkowski et al. (2020). They optimize for a similar objective function to that of variational methods, while adopting architectures from supervised neural optical flow estimation as backbones.

**Robustify Machine Learning Models.** Augmenting the training data with the anticipated shifts and attacks Goodfellow et al. (2015); Kurakin et al. (2017); Madry et al. (2018) is a standard technique for obtaining robust models. However, the robustness decreases when they encounter shifts and attacks the model were not trained on. In addition, adversarial training often leads to significant degradation in clean performance Tsipras et al. (2018). Adversarial training for videos is even more difficult with the growing amount of data and increased dimensionality because it is easier to construct attacks that not showed in training time in a higher dimensional space.

Recently, inference-time defenses have been shown to be effective in improving robustnessKang et al. (2021); Wu et al. (2021); Mao et al. (2021b); Liu et al. (2022); Nie et al. (2022); Tsai et al. (2023), shifting the burden of robustness from training to testing. The model performs robust latent inference by optimizing for specific energy functions Mao et al. (2022c; 2021b); Wang et al. (2019); Tsai et al. (2023). This allows the model to adapt to the unforeseen characteristics of attacks and corruptions at inference time. Since there is no need to retrain the models, inference-time defenses work well with the vast number of available pre-trained models.

**Adversarial Attacks and Defenses on Videos.** Since the discovery of neural networks' vulnerability to adversarial examples, various types of attacks and defenses on static images have been extensively studied. Not until recently did a few studies start investigating adversarial videos. Wei et al. (2019) proposed a perturbation regularized by the $l_{2,1}$ norm that induces sparsity on the adversarial noise. Wei et al. (2020) designed a black-box attack based on heuristically found important frames. Similarly, the attack proposed by Hwang et al. (2021) also relies on the pre-discovery of critical frames in videos, but they take it further to only perturbing one frame in a white-box setting. Pony et al. (2021) on the other hand constructed unconstrained attacks by adding a constant flickering color shift to each frame, obtaining more humanly-imperceptible attacks.

Defense for videos is more challenging and less explored in the literature. Xiao et al. (2019) made the first effort to identify adversarial frames using temporal consistency, without defending them. Jia et al. (2019) proposed a detect-then-defend strategy that classifies the type of attack first and then applies different types of defense accordingly. However, they did not test their defense against any adaptive attacks. Lo & Patel (2021) have a similar adversarial detector component, and proposed to apply multiple independent batch normalization to improve robustness under different types of distribution shifts caused by adversarial perturbations. Their approach requires adversarial training and is only evaluated against weak adversarial attacks.

## 3 MOTION FOR ROBUST PERCEPTION

In this section, we first present a test-time optimization algorithm that respects temporal constraints. We then show how to design the self-supervised objective to capture the temporal constraint efficiently. We lastly show a dense constraint captures the structure better, which boosts the robustness.

### 3.1 ROBUST VIDEO CLASSIFICATION

We consider the family of video perception models using motion, which leads to state-of-the-art performance Carreira & Zisserman (2017) and often obtains better inherent robustness Plizzari et al. (2022); Sevilla-Lara et al. (2018).

Let $X = \{x_1, ... x_T\}$ be the input video clip with $T$ frames, and $Y$ be the label. For every pair of adjacent frames $x_i, x_{i+1}$. The optical flow field $f_i$ is estimated using a pre-trained neural flow estimator $G$: $f_i = G(x_i, x_{i+1})$. Let the neural network that predicts the category using motion to be $H()$. Let the classification loss be $\mathcal{L}()$. We denote the temporal constraint loss as $\mathcal{L}_s()$, which we will discuss in detail later.

---

**Algorithm 1** Robust Inference that Respects Motion

1: **Input:** Potentially corrupted video clip $X$ with $T$ frames, step size $\eta$, number of iterations $K$, flow estimator $G$, video classifier $H$, reverse attack bound $\epsilon_r$.
2: **Output:** Prediction $\hat{y}$
3: **Inference:**
4: $X' \leftarrow X$
5: **for** $k = 1, ..., K$ **do**
6:     $X' \leftarrow X' - \eta \cdot \text{sign}(\nabla_{X'} \mathcal{L}_s)$
7:     $X' \leftarrow \Pi_{(X, \epsilon_r)} X'$    ▷ project back into the bound
8: **end for**
9: Predict the final output by $\hat{y} = H(G(x'))$

---

A standard video classifier will perform inference with the output $\hat{y} = H(G(X))$. We denote the corrupted or attacked video as $X_a$. The attacked video is typically created via:

$$X_a = \arg\max_{X_a} \mathcal{L}(H(G(X_a)), Y)$$

$$s.t. ||X_a - X|| < \epsilon \tag{1}$$

We use optical flow to capture the temporal constraint in videos. For both corrupted videos and attacked videos, we find the estimated flow fields $F$ would also be corrupted (as shown in Figure 1). Note that the attack vector is only optimized to fool the target classifier, yet it also collaterally breaks the temporal constraints in the model.

**Our robust inference:** We will optimize for an additive vector $r$ on the corrupted or attacked video so that it minimizes the temporal constraint loss.

$$r^* = \arg\min_{r} \mathcal{L}_s(G(X_a + r))$$

$$s.t. |r|_p \leq \epsilon_r \tag{2}$$

The reverse perturbation bound $\epsilon_r$ is necessary to avoid trivial solutions. We will run test-time optimization to minimize this temporal consistency loss so that we can repair the corrupted video input. After finding the optimal $r$ through optimization, the video prediction is simply $\hat{y} = H(G(X_a + r))$. Our algorithm is described in Algorithm 1.

### 3.2 Motion Consistency as Temporal Constraint

There are many constraints that a model should respect, such as the constraints for static images like equivariance Mao et al. (2022b). While these methods can be directly applied to videos by treating each individual frame as a static image, they will fail to capture the inconsistency in the temporal dimension.

In this paper, we focus on using motion consistency in videos as a constraint. Motion captures the additional temporal structure in the data, which a robust video perception model should always respect. These constraints coincide with the objective functions of unsupervised optical flow estimation, which has been comprehensively studied in Jonschkowski et al. (2020).

One constraint that video data should respect, is the **brightness consistency**, where the pixel intensity of a scene point does not change when moving through frames Horn & Schunck (1981). If the optical flow is perfectly accurate, we should be able to reconstruct frame $I_t$ by warping frame $I_{t+1}$ with the optical flow between them. We can thus compute a similarity (not limited to $L_p$ distances) between the warped second frame and the first frame as a measurement of flow consistency, which is also commonly named the warping error. We define the photometric loss of a video clip based on brightness consistency by summing over the consistency loss of every adjacent pair of frames.

$$\mathcal{L}_{\text{photo}} = \sum_{t} \mathcal{L}_{\text{sim}}(w(I_{t+1}), I_t) \tag{3}$$

The **smoothness constraint** is another common constraint that natural optical flow satisfies. It regularizes the flow to have a lower spatial frequency and be consistent with neighboring pixels. In this work, we follow Jonschkowski et al. (2020) and define it by the edge-aware second derivative of the estimated flow:

$$\mathcal{L}_{\text{smooth}} = \frac{1}{n} \sum \exp\left(-\frac{\lambda}{3} \sum_{c} |\frac{\partial I_c}{\partial x}|\right) |\frac{\partial^2 V}{\partial x^2}|$$

$$+ \exp\left(-\frac{\lambda}{3} \sum_{c} |\frac{\partial I_c}{\partial y}|\right) |\frac{\partial^2 V}{\partial y^2}| \tag{4}$$

Where $\lambda$ controls the edge weighting, $I_c$ is the pixel intensities of color channel $c$ and $V$ is the estimated optical flow. Given an optical flow estimator $G$ and a similarity metric $\mathcal{L}_{\text{sim}}$, we hereby define the self-supervised temporal constraint as:

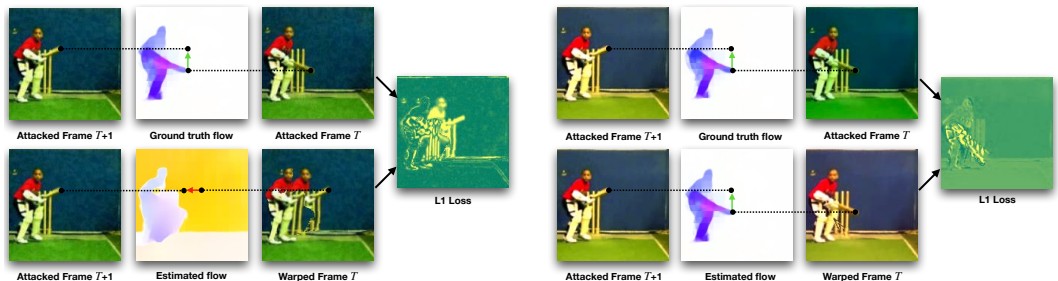

Figure 3: **Our consistency objective can improve robustness in twofold:** (1) it enforces accurate optical flow (left) and (2) it suppresses pixel intensity distortion (right). If the estimated flow is highly corrupted by the attack, as shown on the left, the pixels in the $T + 1$ frame will be warped to the wrong location in frame $T$. The distance between the warped and actual frame $T$ will be high, providing a signal to fix the optical flow. If the attacker shifts the pixel values too far from the original (i.e. under flickering attacksPony et al. (2021)), even if flow fields warp pixels to the correct locations, the pixel distance will still be high. This provides signals to fix discontinuity in the data due to corruption and attacks.

$$\mathcal{L}_{\text{s}}(X, G, \mathcal{L}_{\text{sim}}) = \mathcal{L}_{\text{photo}} + \lambda_{\text{smooth}} \cdot \mathcal{L}_{\text{smooth}} \tag{5}$$

### 3.3 Multiple Motion Constraints For Robustness

It is shown that multitask and multiple constraints Mao et al. (2020); Lawhon et al. (2022); Mao et al. (2022c) are pivotal for adversarial robustness. In Sec. 3.2 we discussed brightness consistency and smoothness as constraints for optical flow restoration. In natural videos, there are rich intrinsic constraints that the model should satisfy. For instance, in previous sections, we used forward flow computation and backward warping (Eq. (3)). We can also compute backward flow with forward warping as an additional constraint.

$$\mathcal{L}_{\text{back}} = \sum_t \mathcal{L}_{\text{sim}}(w_f(I_t), I_{t+1}) \tag{6}$$

In practice, this can be implemented by simply generating flows after reversing the order of the frames. Additionally, motion consistency should continue to hold for longer ranges across time. We can generate flows between frames that are not adjacent, requiring flows to be accurate for larger movements. Here we simply split each clip into $X_{<\frac{T}{2}}$ and $X_{\geq\frac{T}{2}}$, and compute flows between $X_i$ and $X_{i+\frac{T}{2}}$ for $i = 0, ..., \frac{T}{2} - 1$. Therefore, the long-range photometric consistency is defined by:

$$\mathcal{L}_{\text{long}} = \sum_t \mathcal{L}_{\text{sim}}(w(I_t + \frac{T}{2}), I_t) \tag{7}$$

We define the Multiple Motion Consistency objective:

$$\mathcal{L}_{\text{s}}^{\text{multi}} = \mathcal{L}_{\text{forw}} + \mathcal{L}_{\text{back}} + \mathcal{L}_{\text{long}} + \lambda_{\text{smooth}} \cdot \mathcal{L}_{\text{smooth}} \tag{8}$$

In principle, one could continue introducing more constraints, such as photometric consistency between frames for every stride length. However, the computational expense grows linearly with the number of constraints, so in this work, we only consider the constraints above. We show in experiments that this is sufficient to yield improved robustness against defense-aware attacks.

## 4 Experiments

We demonstrate the effectiveness of our defense against various types of corruption and attacks. We first show our model's performance improvement under common natural noise corruptions. We then show its capability of recovering performance under common strong dense attacks. A few attacks designed specifically for videos have been proposed in recent years. We analyze the performance

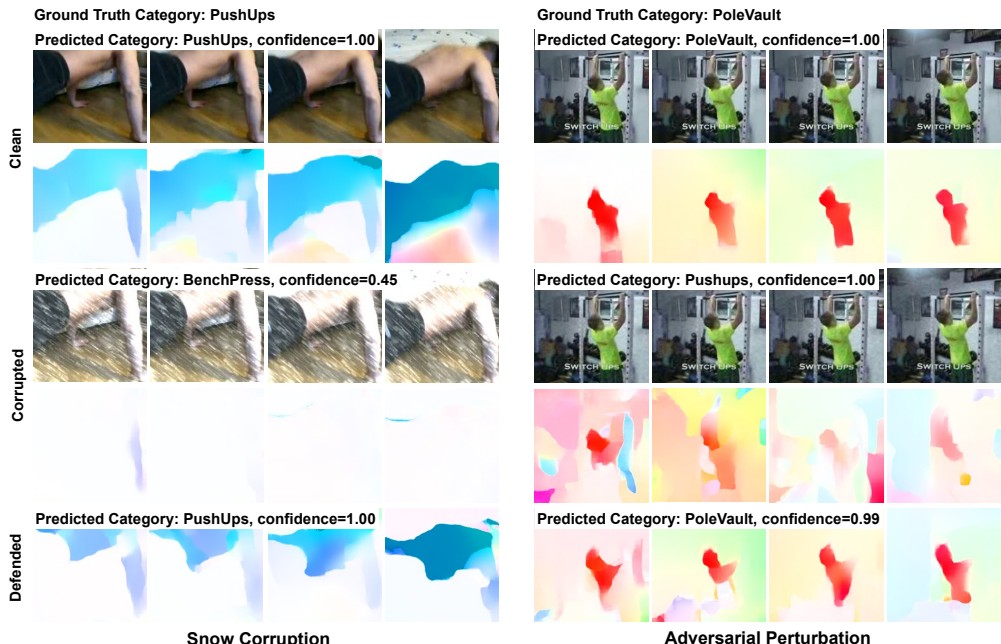

Figure 4: **Visualization of attack and defense.** The left side is a video corrupted by synthetic snow. The right side is a video under adversarial attack. The first and second rows are the RGB frames and estimated flows of the original clean video. The third and fourth rows are RGB frames and flows of corrupted or attacked video. The last row is the optical flow restored by our method. By restoring accurate optical flow, our method also corrects misclassified predictions.

of our defense against two state-of-the-art video attacks as well. Finally, we report results under various defense-aware attacks.

## 4.1 EXPERIMENTAL SETTINGS

**Datasets**. We evaluate the effectiveness of our method on two standard action recognition datasets UCF-101Soomro et al. (2012), and HMDB-51Kuehne et al. (2011). For the natural noise corruptions, we evaluated on the 4 types of noise provided by Schiappa et al. (2023). For the adversarial attack experiments, we evaluated on a subset of 1000 clips for each dataset.

**Implementation Details**. We chose the state-of-the-art RAFTTeed & Deng (2020) as our flow estimator front-end and the flow stream of the state-of-the-art I3DCarreira & Zisserman (2017) as the classifier backbone. As reported by the original authors, RAFT has strong generalization across datasets. We directly use the RAFT model pre-trained on the Sintel dataset Butler et al. (2012), as it was already capable of producing high-quality flows on UCF-101. We then fine-tune I3D models pre-trained on Kinetics Carreira & Zisserman (2017) to our datasets. We resize all videos to $300 \times 400$ pixels and center crop $224 \times 224$ to eliminate black surrounding frames. We randomly sample 64-frame clips from videos. It is worth noting that we evaluate on videos with the largest input dimensions compared to prior works, which is more difficult to defend.

We found that using simple $L_1$ distance as the similarity metric in Eq. (3) was sufficient to yield effective results. Details on the selection of similarity metrics can be found in the appendix.

## 4.2 ROBUSTNESS AGAINST NATURAL CORRUPTIONS

From Tab. 1 we can see that our defense is effective across 4 types of common noise corruption. Our defense improves performance by up to 6 points. We also provide analysis of various other types of common corruptions in the appendix and show a correlation between the effectiveness of our defense and the magnitude of flow change caused by the corruption.

| Noise | Standard | Defended |
|---|---|---|
| Gaussian | 20.43 | **26.06** |
| Shot | 22.63 | **26.51** |
| Impulse | 20.09 | **26.57** |
| Speckle | 21.25 | **26.57** |

Table 1: Robustness accuracies under 4 natural noise corruptions from UCF101-P

## 4.3 ROBUSTNESS AGAINST ADVERSARIAL ATTACKS

**Dense Attacks.** Projected Gradient Descent (PGD) is a commonly applied strong dense attack. It finds worst-case perturbations by randomly initializing in the given bound and iteratively updating the attack vector, projecting it back into the bound whenever it is exceeded. We view video clips as 4-D tensors in our experiments and deploy PGD with various $L_\infty$ bounds, corresponding to different attack strengths. We applied 20 steps for all PGD attacks. We also evaluate against AutoAttack Croce & Hein (2020), a state-of-the-art parameter-free dense attack combining different objectives. In our experiments, we applied AutoPGD with a combination of cross-entropy and Difference of Logits Ratio (DLR) loss. We found 10 iterations were sufficiently strong. For our defense, we apply $K = 20$ steps of iterations with step size $\eta = 2$ and $L_\infty$ bound $\epsilon_r = 12$.

| Model | UCF-101 | | | | HMDB-51 | | | |
|---|---|---|---|---|---|---|---|---|
| | Standard | Random | Defended | Defended (Multi) | Standard | Random | Defended | Defended (Multi) |
| Clean | 86.6 | - | 84.0 | - | 67.0 | - | 66.9 | - |
| Random | 79.9 | - | 82.2 | - | 62.7 | - | 63.9 | - |
| PGD ($\epsilon = 4$) | 14.7 | 15.3 | **78.5** | 77.9 | 8.8 | 8.8 | 59.5 | **60.1** |
| PGD ($\epsilon = 8$) | 10.5 | 10.9 | **73.9** | 73.8 | 7.9 | 7.7 | **53.9** | 53.6 |
| PGD ($\epsilon = 16$) | 6.2 | 6.7 | **54.5** | 52.8 | 5.2 | 6.6 | 40.8 | **41.4** |
| AutoAttack ($\epsilon = 8$) | 2.4 | 6.9 | **75.3** | 73.4 | 1.3 | 5.6 | 57.9 | **60.2** |

Table 2: Adversarial robust accuracies against dense attacks on UCF-101 and HMDB-51. By restoring the intrinsic motion information, our defense consistently improves performance under various types and strengths of dense attacks. Our method also preserves clean accuracies. With a cost of fewer than 3 points drop in clean accuracy, we obtain a robustness gain by up to 72 points.

Quantitative results on defending dense attacks are reported in Tab. 2. For comparison, "random" is applying uniform noise with the same bound as the attack/defense. Our defense is effective under various types and strengths of attacks. Clean accuracy is also preserved to a considerable degree. We show visualizations in Fig. 4. Dense attacks in the RGB space result in corrupted optical flow estimations and misclassified labels. After our defense, accurate optical flow is restored and classification results are corrected.

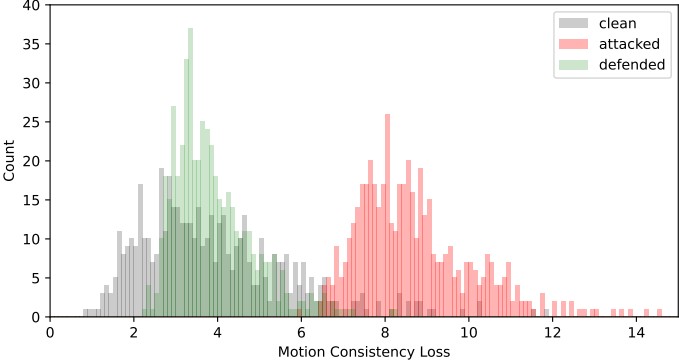

Figure 5: **Motion Consistency Loss distribution.** The x axis is the Motion Consistency loss value, defined in Equation 3, the y axis is frequency. Attacked videos (red) have significantly higher losses than clean ones (gray), whereas the defended videos (green) are pushed back to a lower loss distribution.

We can further plot the distribution of motion consistency losses over the test set videos for different examples. As shown in Fig. 5, clean videos have relatively low losses, indicating that optical flow estimated from natural videos are consistent. Attacked videos have significantly higher losses, suggesting the motion consistency has been broken. Defended videos have lower losses, showing motion consistency has been restored.

**One-Frame Attacks.** The one-frame attack introduced in Hwang et al. (2021) suggests that an attacker can identify the 'weak' frame in a video clip and add perturbations only to that frame to achieve success in fooling the model. While PGD attacks can be seen as the worst-case perturbation for an $L_p$ constraint, one-frame attacks are weaker since additional constraints are added to the attacker, trading off attack strength for imperceptibility. We follow the original paper's evaluation protocol of using 32-frame clips, as we found that one-frame attacks were much less effective on 64-frame clips. Results in Tab. 3 show the effectiveness of our defense. We also find that one-frame attacks are fragile as the fooling rate can be greatly reduced by simply adding uniform noise on

attacked videos. Moreover, we observe a natural resistance toward one-frame attacks in our model even without defense.

| | UCF-101 | | |
|---|---|---|---|
| Model | Standard | Random | Defended |
| Clean | 79.7 | 71.4 | 77.5 |
| One-frame | 34.3 | 59.1 | **76.6** |

Table 3: Adversarial robust accuracies against one-frame attacks on UCF-101. Our defense improves robustness by up to 42 points.

| | UCF-101 | |
|---|---|---|
| Model | Standard | Defended |
| Clean | 84.0 | 84.0 |
| Flickering | 34.0 | **79.5** |

Table 4: Adversarial robust accuracies against one-frame attacks on UCF-101. Our defense improves robustness by up to 42 points.

**Over-the-air Flickering Attacks.** The authors of Pony et al. (2021) described an attack that only applies a constant color shift to each individual frame of a video. It is not constrained to an $L_p$ bound and aims at being imperceptible to human eyes by regularizing the magnitude and frequency of change (thickness and roughness) of the adversarial perturbations.

We follow the original paper and sample 90-frame clips. Flickering attacks are computationally expensive as an attack on each clip takes 1000 iterations. To reduce computational cost, we only perform 300 steps. Note that this makes the attack harder to defend as the perturbation has larger magnitudes, often exceeding our defense bound. Nevertheless, our defense remains effective. For flickering attacks, we evaluated on a 200-clip subset.

Interestingly, as shown in Fig. 6, we find that the flow fields were not significantly distorted. This is because the relative pixel values are not changed and pattern correspondence is still easy to find. However, the motion consistency loss is still much larger than in clean videos since there are shifts in pixel values between frames. As illustrated in Fig. 3, even if an accurate flow learns to warp perfectly, there will still be a constant color difference between the true first frame and the reconstructed one. To minimize this difference, our defense learns to suppress the color shift, resulting in defended videos. In this case, the defended flows even exhibit better qualities than clean ones.

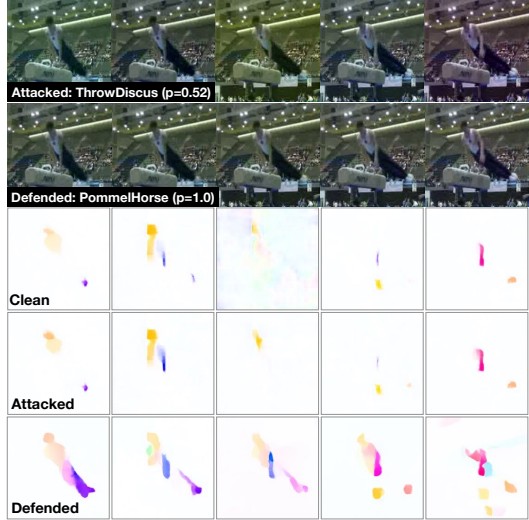

Figure 6: **Resisting Flickering Attacks.** Flickering attacks fool the video action classifier by shifting the color, which is beyond the $L_\infty$ bounded attack. Row 1 is the RGB frames of a "PommelHorse" video under flickering attack. Row 2 is the defended frames. Row 3, 4, 5 are the flow fields estimated on clean, attacked and defended inputs, respectively. While flickering attacks do not significantly distort estimated flows, our defense is still effective because it suppresses the color shift in the inputs to lower consistency losses, which counters the adversarial attack.

## 4.4 ANALYSIS: DEFENDING AGAINST ADAPTIVE ATTACKS

If the attacker had knowledge of the defense deployed, they can adjust their strategy accordingly. We discuss three types of defense-aware attacks targeting our model.

**Adaptive Attack I.** One way a defense-aware attacker could make their attack more effective is to respect the defender's loss in their attack objective. As described in Mao et al. (2021b), one could construct an attack that maximizes the cross-entropy loss while minimizing the defender's loss at the same time by introducing a Lagrangian multiplier.

$$\delta_{\mathbf{AA1}} = \arg\max_\delta \mathcal{L} - \lambda \mathcal{L}_s \tag{9}$$

By incorporating the defender's loss as a regularization, the attacker finds examples that are more likely to circumvent the defense.

**Adaptive Attack II.** The attack described above is a general way to counter defenses. We design another attack that specifically aims at bypassing our defense. Recall the observation in Fig. 1. The assumption that our defense relies on is that adversarial perturbations in the RGB space result in corrupted motion estimation. If an attacker was aware of our defense, they could apply attacks has minimal effects on the estimated motion by adding the change of flow as a regularization term.

$$\delta_{\mathbf{AA2}} = \arg\max_{\delta} \mathcal{L} - \lambda |G(x + \delta) - G(x)|_p \quad (10)$$

By doing so, the attack becomes more "stealthy" and provides a weaker signal for our defense.

**Adaptive Attack III.** A common issue for inference-time defenses is the reliance on obfuscated gradients. As pointed out in Croce et al. (2022), the iterative process for purification potentially leads to vanishing gradients, which provides a false sense of security as the attacker can deploy gradient approximation attacks, known as BDPA Athalye et al. (2018), to circumvent this. In our experiments, we follow Croce et al. (2022), applying full forward pass for the purification, and replacing its gradients with the identity function during backward pass.

Evaluation of our defense against three adaptive attacks is summarized in Tab. 5. All attacks are $L_\infty = 8$ bounded and defenses are $L_\infty = 12$ bounded. All attacks and defenses are obtained through 20 iterations. Adaptive attacks are generally computationally intensive, experiments in this section are evaluated on a 200-clip subset of the test set. We can see that compared to static attacks, adaptive attacks are stronger as the defended accuracies are lowered to various extents. Notably, BPDA is the strongest attack.

| Model | UCF-101 | | |
|---|---|---|---|
| | Standard | Defended | Defended (Multi) |
| Clean | 85.5 | 83.5 | 81.0 |
| PGD* | 10.5 | 73.9 | - |
| Adaptive I | 28.0 | 65.0 | **68.5** |
| Adaptive II (FlowReg) | 64.5 | 68.5 | **72.5** |
| Adaptive III (BPDA) | 18.5 | 20.5 | **25.5** |

Table 5: Adversarial robust accuracies under adaptive attacks on UCF-101 dataset. We show robust accuracy under three types of adaptive attacks. For reference, we also show the standard PGD attack, where the * indicates that the attack is evaluated on 1000 examples. By applying our defense with multiple motion constraints, we can obtain a robust accuracy of 25.5, which is over 15 points higher than the standard defense attacked by PGD. Our defense also improves BPDA attacked videos by 7 points. Additionally, using multiple constraints consistently improves robustness.

We also observe that our multiple constraint defense consistently improved robustness. This is because when multiple constraints are introduced, the attacker has to solve for a multi-task optimization, which is inherently hard due to more constraints. The attack budget has to be spent not only on fooling the classifier but also on satisfying the various constraints. Another consequence of this is weaker effectiveness of the attacks on standard models. The same lose-lose situation for the attacker described in Mao et al. (2021b) is observed here, namely if the attacker chooses to take our defense into consideration, the effectiveness on undefended models is weakened; if not, our defense continues to be strongly effective.

## 5 CONCLUSIONS

In this work we show that natural corruptions and adversarial attacks on video classification not only fool the classifier but also disrupt the motion consistency of the videos. We proposed a novel inference-time defense, improving robustness by restoring motion via self-supervised optical flow consistency. Empirical results show robustness under a variety of corruptions and attacks. Our work is the first inference-time defense for videos that uses motion consistency to improve robustness. Our defense is effective even under strong adaptive attacks. Our work suggests that video perception models are fragile partially because they ignore the rich motion structure in the natural data.

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

# A APPENDIX

## A.1 VISUALIZATIONS

### A.1.1 RESTORING FLOWS

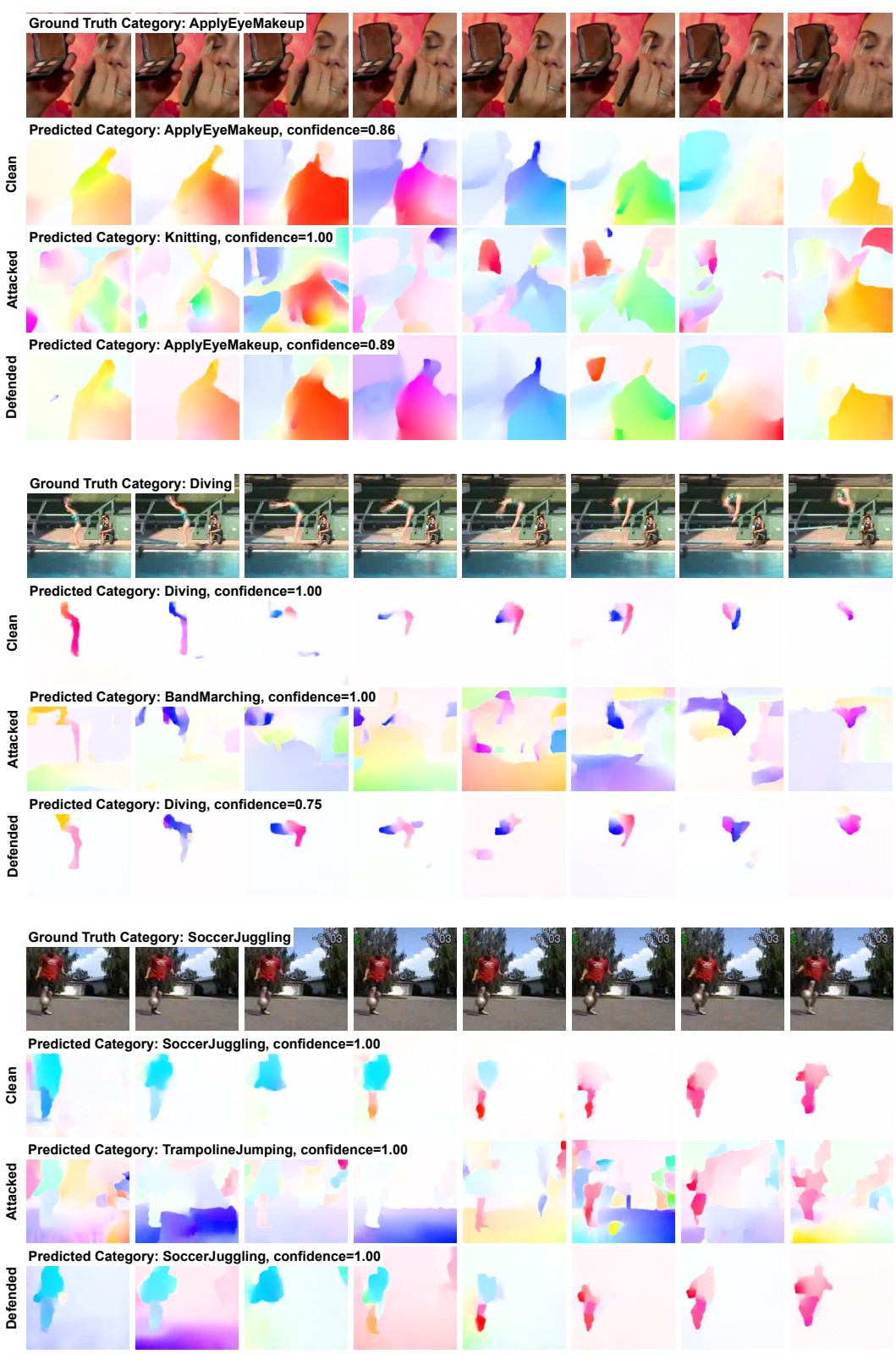

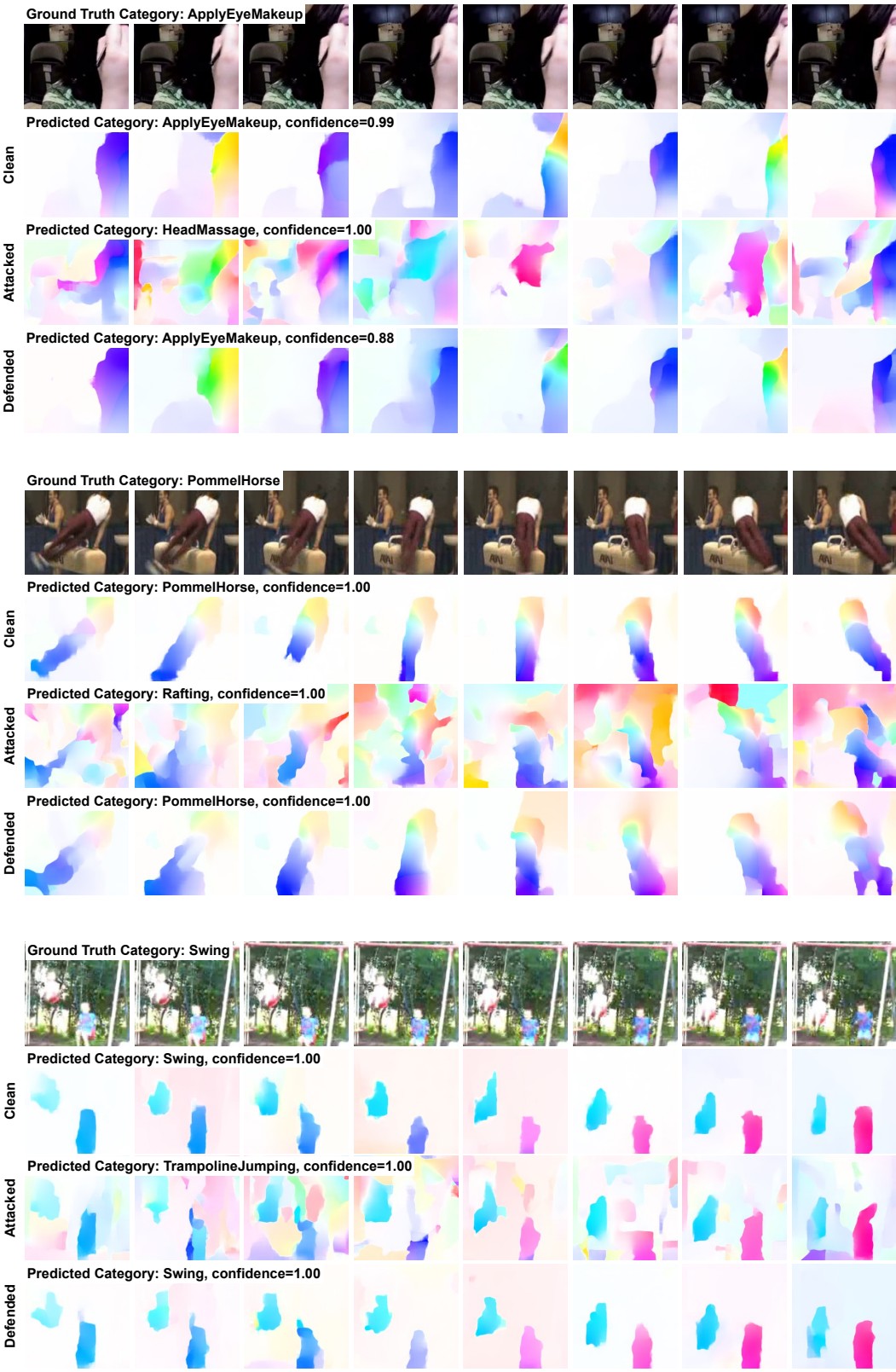

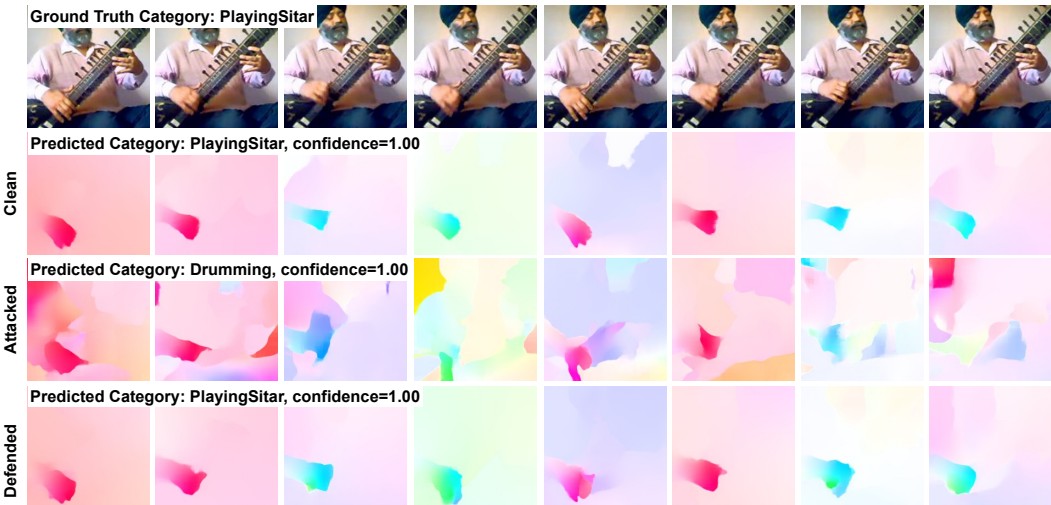

Figure 7: Visualization of adversarial attack and defense. The first rows are the frames of the original video. Row 2, 3 and 4 are the optical flow fields estimated on clean, attacked, and defended inputs, respectively. The adversarial attack on RGB inputs resulted in the wrong classification prediction as well as corrupted flow estimations. Using our defense method, we restore the accurate flow fields as well as the correct categorical prediction of the model.

### A.1.2 WARPED VIDEOS

We include visualizations of warped videos using clean, attacked and defended videos in the uploaded zip file. Every frame in these videos is warped from the ground truth next frame using the optical flow estimated between them. The attacked videos have inaccurate estimated flows and result in visibly distorted warped videos. The optical flow is repaired in defended videos, restoring the motion consistency and correcting classification results.

## A.2 RESULTS

### A.2.1 OTHER NATURAL COMMON CORRUPTIONS

We evaluate our method on 2 sets of synthetic common corruptions. One set is adapted from corruptions used in Imagenet-C Hendrycks & Dietterich (2019), where static corruption is applied to every frame independently. To better mimic real-world corruptions on videos, we also adapted the Image-P-like corruptions Hendrycks & Dietterich (2019), where temporal-related corruptions are sequentially added to each frame. To better target motion-related misclassification, the defense vector is only applied if the motion constraint loss is decreased over 10% after optimization. Frame-by-frame corruption synthesis is expensive so we evaluated on a 200 clip subset of the test set. Results in Tab. 6 and Tab. 7 show that our method is effective for a majority of the corruptions, with an average improvement of 1.7 and 1.2 points. As shown in Fig. 8, across all types of corruption, we find that there is a 0.46 Pearson correlation between the average motion consistency loss of a type of corruption and the accuracy change after deploying our defense. This indicates that our defense is more effective on the corruptions that disrupt motion consistency.

| Model | Clean | Noise | | | Blur | | | | Weather | | | | Digital | | | | Avg |
|---|---|---|---|---|---|---|---|---|---|---|---|---|---|---|---|---|---|
| | | Gauss. | Shot | Impul | Defoc | Glass | Motion | Zoom | Snow | Frost | Fog | Bright | Cont | Elast | Pixel | JPEG | |
| Standard | 88.5 | 26.0 | 31.5 | 26.5 | 78.5 | 56.0 | 73.5 | 82.0 | 21.0 | 10.0 | 72.5 | 85.5 | 85.0 | 40.5 | 89.0 | 69.0 | 56.4 |
| Ours | **90.5** | **31.5** | **33.5** | **56.0** | 74.5 | **67.0** | 60.5 | 78.5 | **34.0** | **16.0** | **77.0** | **87.5** | 69.0 | 29.5 | 77.5 | **79.0** | **58.1** |

Table 6: Robustness evaluation on Imagenet-C-like corruptions. Our method improves performance under most corruptions with an average improvement of 1.7 points in accuracy.

| Model | Clean | Noise | | Blur | | Weather | | Digital | | | | Avg |
| | | Gaussian | Shot | Motion | Zoom | Snow | Bright | Translate | Rotate | Tilt | Scale | |
|---|---|---|---|---|---|---|---|---|---|---|---|---|
| Standard | 88.5 | 68.0 | 74.5 | 78.5 | 40.5 | 66.0 | 83.0 | 84.5 | 68.0 | 65.0 | 66.5 | 69.5 |
| Ours | **90.5** | **70.5** | **78.5** | **80.0** | **45.5** | **71.0** | 79.5 | 81.5 | **70.0** | 64.0 | 66.0 | **70.7** |

Table 7: Robustness evaluation on Imagenet-P-like corruptions. Our method improves performance under most corruptions with an average improvement of 1.2 points in accuracy.

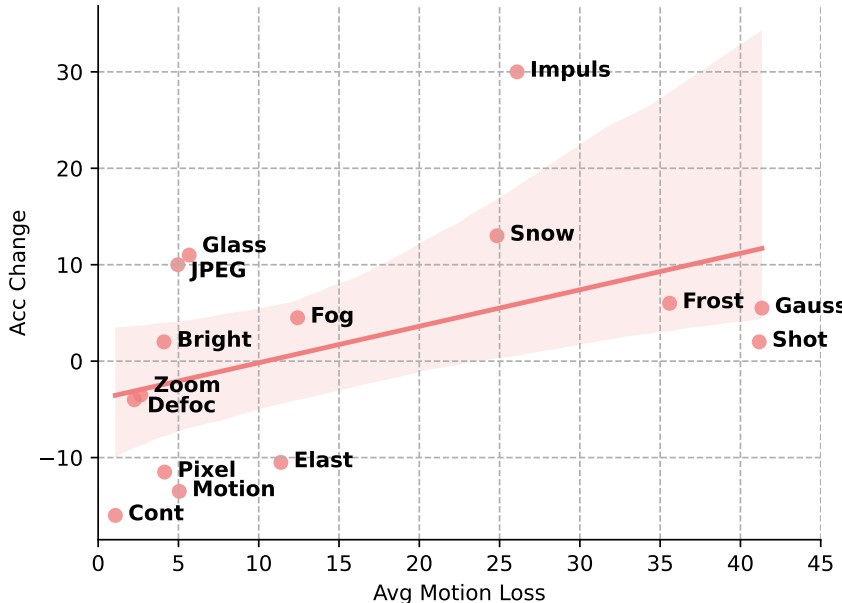

Figure 8: **Correlation between motion consistency loss and accuracy change after defense.** Our defense is more effective on corruptions that have larger disruptions in motion.

### A.2.2 ADDITIONAL BASELINES

In this section, we report our method's performance against the two test-time adaptive defenses proposed in Mao et al. (2022c), which are based on equivariance and invariance constraints. We show that our defense based on motion consistency constraints has superior performance over other constraints for videos.

| model | Acc |
|---|---|
| Clean | 79.7 |
| PGD | 10.2 |
| InvarianceMao et al. (2022c) | 24.7 |
| EquivarianceMao et al. (2022c) | 25.9 |
| Motion (Ours) | **68.2** |

Table 8: Baselines

Due to memory and backbone architecture limitations, we evaluated on 32-frame clips for all models in this section; for equivariance and invariance, we used 4 transformations: 2 rotations, 1 color jitter and 1 horizontal flip.

### A.2.3 RAFT RECURRENT STEPS

The GRUCho et al. (2014) module in RAFT Teed & Deng (2020) has a default setting of 12 recurrent steps. When directly applying this setting, we find that the undefended model has a natural robust accuracy of above 20% under end-to-end $L_\infty = 8$ PGD attacks. While this could be a benefit of optical flow's invariance to appearance, we also investigate the possibility of reliance on obfuscated gradients. It is well-known that recurrent neural networks are prone to vanishing and exploding gradients Bengio et al. (1994). We examine this possibility by approximating the gradients of the

GRU during attack. Specifically, we simply reduce the number of recurrent steps when computing gradients, then perform full 12-step forward pass during actual inference. Robust accuracies under various attack budgets using different recurrent iters are plotted in Fig. 9.

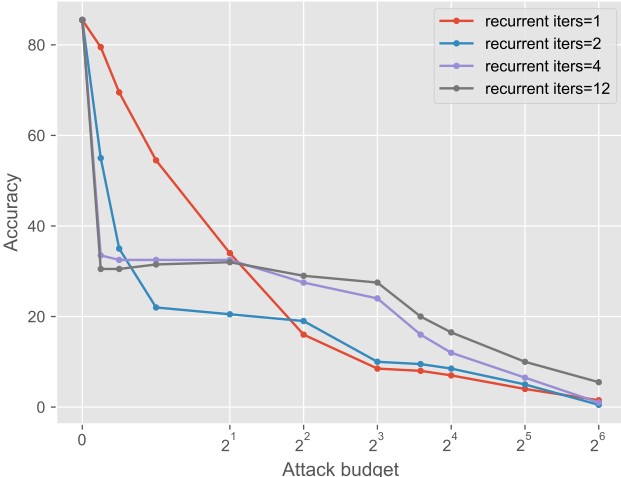

Figure 9: Accuracies under PGD attacks of RAFT using different iters in the recurrent module. The x axis is the attack budget and the y axis is the accuracy of models. Every curve corresponds to a model with a specific number of recurrent steps in the GRU module when computing gradients. Note that after the attack is generated, all models still perform full 12 recurrent steps at final inference. We find that for larger attack budgets, using less recurrent steps to approximate gradients results in reduced accuracy. If the number of recurrent steps is too small, the gradient approximation becomes inaccurate, also rendering attack ineffective. We balance this by choosing recurrent iters=2.

We find that for attacks with large budgets, the natural undefended accuracy can be reduced by approximating gradients, indicating a certain extent of reliance on obfuscated gradients when applying full 12 recurrent steps. However, when the number of step is too small, the gradients are not an accurate approximation of the inference time 12-step gradients. We chose recurrent iters = 2 as it balances the two: remains a good approximation of the 12-step forward pass gradients while keeping the reliance on obfuscated gradients at minimum. This baseline has a reduced undefended accuracy of 10.5%.

### A.2.4 SIMILARITY METRICS

In Section 3.2 of the paper, we described using a similarity metric between warped and ground truth frames as the photometric consistency loss. As the authors of Jonschkowski et al. (2020) have discussed, there are many choices for this metric. We experimented on the $L_1$, $L_2$, and $SSIM$ distances. Robust accuracies using only photometric loss with different similarity metrics are reported in Tab. 10. In terms of improving defense effectiveness, $L_1$ is better than or same as other metrics so for this work we used the simplest $L_1$ distance for photometric consistency.

| metric | Defended Acc |
|--------|--------------|
| $L_1$ | 66.5 |
| $L_2$ | 61.5 |
| SSIM | 66.5 |

Table 9: Metrics

### A.2.5 HYPERPARAMETER OF ADAPTIVE ATTACKS

In section 4.3 of our paper, we described our adaptive attacks. For Adaptive Attack I and Adaptive Attack II (FlowReg) we followed **?**, applying the attack by solving a Lagrangian that incorporates a

defend-aware term:

$$\delta_{\mathbf{AA1}} = \arg\max_{\delta} \mathcal{L}_{\text{CE}} - \lambda_1 \mathcal{L}_{\mathcal{MC}}$$

$$\delta_{\mathbf{AA2}} = \arg\max_{\delta} \mathcal{L}_{\text{CE}} - \lambda_2 |G(x+\delta) - G(x)|_p \tag{11}$$

We perform a parameter sweep of $\lambda$ to find the strongest attacks. This is shown in Fig. 10

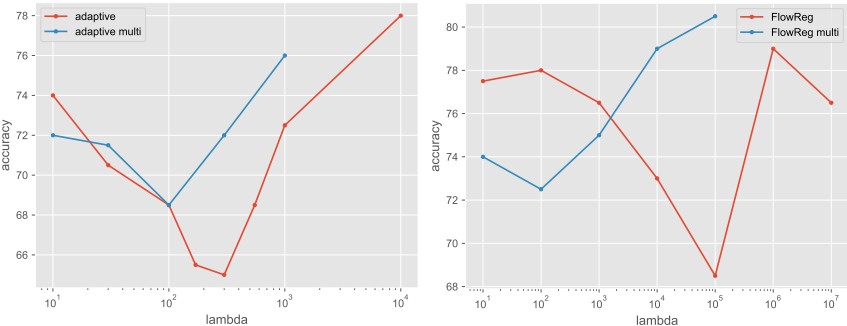

Figure 10: Parameter sweep for the strongest adaptive attacks. In both plots the x axis is the value of $\lambda$, the y axis is the defended accuracy of each adaptive attack. Lower is better, indicating that the attack is more capable of circumventing the defense. Red curves are using the baseline defense, blue curves are using multiple-constraints defense. The left figure is Adaptive Attack I, where the optimal $\lambda = 300$. The right is Adaptive Attack II, where the optimal $\lambda = 10^5$

A.2.6 DEFENSE STEPS AND CLIP LENGTH

In Fig. 11, we report the effect of defense steps and clip lengths.

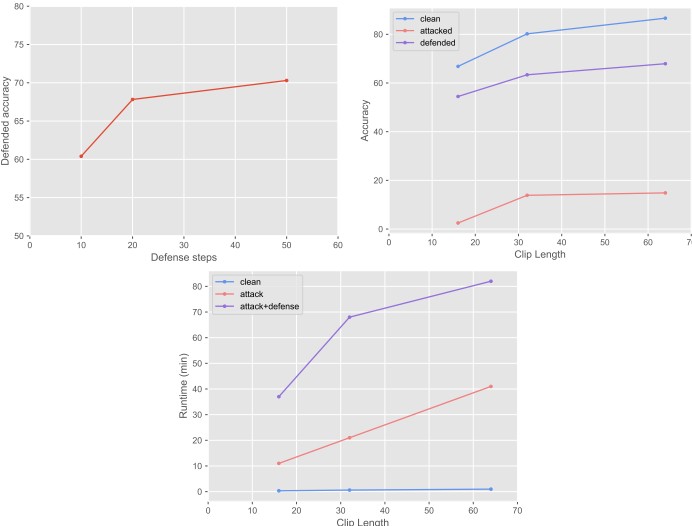

Figure 11: Effect of defense steps and clip Length. In the figure on the left, we show how defended accuracy is effected by the change in number of defense steps. We find 20 steps to be sufficient for high effectiveness. In the figure in the middle, we show how the length of clips effect clean, attacked, and defended accuracies. In the figure on the right, we show how the length of clips effect the runtime of inference.

## A.3 IMPLEMENTATION DETAILS

### A.3.1 EFFICIENT IMPLEMENTATION OF RAFT FOR VIDEOS

The original RAFT Teed & Deng (2020) was implemented for computing optical flow between one pair of frames. They first use an encoder to extract features per-pixel. This is inefficient when computing stacks of flow fields for a video clip with multiple pair of frames, as all frames except for the first and the last will be passed through the encoder twice, once as the source frame and once as the target frame. We modified the implementation by pre-computing visual features of every frame first, and using them when needed. This resulted in a faster and memory efficient implementation for video clips.

|  | Original | Efficient |
|---|---|---|
| Runtime (s) | 23.27 | **17.61** |
| GPU Memory (MB) | 30,888 | **27,056** |

Table 10: Efficient implementation of RAFT for videos. . Results are reported performing one forward pass of a 64-frame clip. Our efficient implementation reduces runtime and memory consumption.

### A.3.2 OTHER MOTION CONSISTENCY CONSCTRAINTS

Unsupervised optical flow learning has a rich literature, providing us with many handy tricks for measuring optical flow quality. In our work, we only experimented with the basic photometric consistency and smoothness constraints. It is natural to extend our method to including more flow quality assessment techniques, for example, occlusion-aware consistency. We leave that to future work.

Note that the recurrent update module in RAFT has the potential to create obfuscated gradients, which has been shown to lead to a false sense of security Athalye et al. (2018). We circumvent this by approximating gradients during attack and defense. Specifically, the recurrent module only runs 2 iterations during gradient computation in attack and defense, allowing us to obtain useful gradients that approximate the true ones while keeping the risk of exploding and vanishing gradients at minimum. At actual inference time, the number of updates is switched back to the default 12 iterations. A more detailed study is included in the appendix.

We found that using simple $L_1$ distance as the similarity metric in Eq. (3) was sufficient to yield effective results. Details on the selection of similarity metrics can be found in the appendix.

