# OpenReview forum: "Robust Video Perception by Seeing Motion"
_ICLR.cc/2024/Conference — ICLR 2024 Conference Withdrawn Submission_

### Official Review · Reviewer_AKHJ · 2023-10-25

**Soundness:** 3 good
**Presentation:** 3 good
**Contribution:** 2 fair
**Rating:** 5
**Confidence:** 3

**Summary:**

The core idea of the paper is to use motion as a temporal constraint to improve the robustness of video perception tasks. The paper presents a test-time optimization algorithm using the temporal constraint and tests which is then evaluated on UCF101 and HMDB-51 by conducting a video classification task.

**Strengths:**

1. Using motion as a constraint for video defense is intuitive and effective for certain types of attacks.
2. The proposed test-time optimization is intriguing.
3. The analysis section is resourceful.
4. The paper is easy to follow.

**Weaknesses:**

1. The main concern is the assumption of the paper may be too strong. Authors assume all the attacks will destroy the optical flow and that the flows can be recovered given the existing information. In Figure 5, the authors demonstrate how the motion consistency is different in clean, attacked, and defended videos. However, this may not be a general situation, for example, the rectangular occlusion attack (ROA) mentioned in [1]. There are many other types of common video attacks that have not been discussed in the paper.
2. I understand that the proposed method is a test-time optimization method, but it can not be an excuse to not compare with other relative existing methods in the paper.
3. Necessary ablations are missing. There are hyperparameters like step size, K, and bounds but the authors did not mention how sensitive is the framework towards those hyperparameters.

[1] Wu, T., Tong, L., & Vorobeychik, Y. (2019). Defending against physically realizable attacks on image classification. arXiv preprint arXiv:1909.09552.

**Questions:**

1. How long will it take for test-time optimization to defend each type of attack?
2. During the experiments, have authors ever encountered certain types of attacks that were not very suitable for the proposed defense methods?

---

### Official Review · Reviewer_NZbs · 2023-10-29

**Soundness:** 2 fair
**Presentation:** 2 fair
**Contribution:** 2 fair
**Rating:** 3
**Confidence:** 3

**Summary:**

The authors propose a method to help video perception by fixing the potential corruption of video signals. The fix replies on the shift reflected in the motion flows. The idea is to improve the video perception by adopting test-time video repairing. The proposed method is claimed to be effective to both corrupted and attacked (under certain assumptions) videos.

**Strengths:**

- The proposed method is supported with the very intuitive idea and straightforward implementations that is easy to follow.
- The writing is basically easy-to-follow. The technical section can be quick to understand even to someone who not actively working in this domain.
- The experiments provide empirical evidence to support the effectiveness of the proposed method under different protocols and on different datasets.

**Weaknesses:**

- The experiments lack comparison with other existing solutions in this line. This makes the estimation of method significance hard.
- The proposed methods rely on the multiple terms of constraints, such as smoothness, brightness etc. However, all these constraints are studied for long in previous works. I thus have no confidence about the novelty of the proposed method in this paper.
- In the related works, the authors discuss the motion estimation methods, “Robustify Machine Learning Models.” (I believe “robustify” is a typo here) and “Adversarial Attacks and Defenses on Videos”. However, to someone not familiar with the domain, it is expected to discuss whether the used techniques in the method such as the involved consistency terms and the idea of “repairing then perception” has been adopted in related works. I would expect the authors to make more clear claim and elaboration about the novelty of the proposed method.

**Questions:**

Overall, my concern is on the novelty of the proposed method and its empirical evidence. I would expect more elaboration on the novelty and a comparison with related works published recently to help calibrate the experimental significance.

---

### Official Review · Reviewer_rbXC · 2023-10-31

**Soundness:** 2 fair
**Presentation:** 3 good
**Contribution:** 3 good
**Rating:** 3
**Confidence:** 3

**Summary:**

This paper proposes a motion-based method for improving the robustness of video
classification models to adversarial pertubations and natural corruptions such as
noise or challenging wheather conditions. The base model first computes optical
flow and then classifies the flow field using an established architecture. As a defense,
the authors optimize a small change to the input images in order to minimize the
warping loss as used for training unsupervised optical flow estimation methods. The
video is then classified using the adapted images with the original model. This
test-time optimization is shown to improve the robustness of the model for several
attack strategies.

**Strengths:**

- Robustness to noise and adversarial pertubations is a very relevant topic. Explicitely
  using the motion in videos to improve robustness is a plausible and interesting
  approach.
- The paper is well written and therefore easy to follow.

**Weaknesses:**

In my view, the main weakness of the paper is the empirical evaluation of the
adversarial robustness of the model. However I do not have a background in adversarial
robustness and would therefore be interested in the other reviewers' opinions about
these points.

- The evaluation in section 4.3 seems to be irrelevant to me. The adversarial pertubation
  is optimized using the original, undefended model but tested on the adapted, defended
  model. From a high performance on these stimuli we do not learn anything about the
  robustness of the adapted model.
- In section 4.4, attacks against the adapted model are considered.
    - The first two attacks use knowledge of the test time loss in the optimization of
      the adversarial pertubation. The additional constraints however seem to be
      limiting. For example, an adversarial pertubation that hardly changes the optical
      flow estimate as optimized in "Adaptive Attack 2" will clearly be problematic for
      the defense. However, adversarial pertubations with this property are only a
      subset of all adversarial pertubations. There might be strong adversarial
      pertubations that cannot be found with this approach.
    - Using gradient approximation as in Adaptive Attack 3 is a much stronger approach.
      Consequently, much stronger pertubations are found in which only a small fraction
      of the performance is restored by the defense (Table 5 last row). As mentioned by
      the authors, the better performance of the multiple constraints defense seems to
      mainly stem from the computational cost of attacks.
- Only gradient-based attacks are considered in this paper. Since gradient-based attacks
  are computationally difficult for a defense relying on test-time optimization, it
  would be interesting to also consider a gradient-free attack
  (cf. [Carlini et al. 2019](https://arxiv.org/pdf/1902.06705.pdf)).

In summary, I am not convinced that the adapted model is adversarially substantially
more robust than the base model. The good results reported for many attacks rather seem
to be due to the computational cd of optimizing adversarial pertubations due to the
use of test-time optimization.

**Questions:**

- What is the clean performance for the results reported in Table 1? Is it 86.6 as
  reported in Table 2? If so, the defense seems to only restore model performance to a
  small degree. Why is this the case?

---

### Official Review · Reviewer_EksZ · 2023-11-01

**Soundness:** 2 fair
**Presentation:** 3 good
**Contribution:** 2 fair
**Rating:** 5
**Confidence:** 2

**Summary:**

In this paper, the authors find that natural corruption and adversarial attacks harm both video classifiers and motion estimation. Thus they try to improve the model's robustness by a test-time constraint using motion information. Experiments on UCF and HMDB demonstrate its effectiveness.

**Strengths:**

- The paper is well-written and organized, with clear figures and tables.
- The logic is clear and easy to lead.
- As the authors claim, `this is the first inference-time defense for videos that uses motion consistency to improve robustness.`

**Weaknesses:**

As I'm not familiar with this research topic, I may not give a fair review:
1. What does the `standard` mean in different tables?
2. How about the time consuming? Is it better than those methods that need training?
3. Is this method suitable for those with videos with quick motion?
4. How does this method extend to general action recognition, since most of the current video backbones only use RGB frames?

**Questions:**

- The reference style in the main paper may be wrong. `Mao et al. (2020)` should be `(Mao et al., 2020)` for ICLR.
- The caption for Table 4 is wrong.

---

### Official Review · Reviewer_Zd57 · 2023-11-01

**Soundness:** 3 good
**Presentation:** 3 good
**Contribution:** 3 good
**Rating:** 5
**Confidence:** 4

**Summary:**

This paper proposes a solution based on the recovery of action information to address the issues of distribution shift and adversarial attacks in computer vision models, thereby enhancing the accuracy of the model during testing. The paper, from the perspective of action information consistency, designs multiple constraints to recover action information and validates the method on two datasets. Experimental results show no significant improvement under noisy conditions but demonstrate satisfactory results under adversarial attacks.

**Strengths:**

- This paper conducts experiments using various types of adversarial attacks, and the proposed method shows clear improvements.
- The design of the adversarial attacks is innovative. The paper also discusses the impact of adaptive adversarial attacks, and even under strong adaptive attacks, the proposed method still performs well.
- The proposed method is relatively simple and can be applied at the testing phase. It can be used in conjunction with other methods to enhance model robustness. Depending on the need, one can choose whether or not to use this method.

**Weaknesses:**

- This paper investigates the issue of robustness in video action recognition, but it lacks comparison with test-time adaptation (TTA) methods, such as [A-B]. These TTA methods also aim to adapt to out-of-distribution data when the input data is disturbed by noise. Although these TTA methods mainly focus on updating model parameters, and this paper primarily focuses on adjusting the input data, how to prove that data processing is superior to model parameter adjustment? I believe a comparison should be made based on experimental results.
- Under noisy conditions, many TTA methods can achieve desirable results, while the improvement brought by this paper's method is relatively low.
- In appendix A.2.1, under noisy conditions, the average performance improvement brought by this paper's method is very low and can even be counterproductive under certain noise conditions. Does this indicate an issue with the approach of changing input data?
- How to verify the reliability of the long-range photometric consistency in section 3.3? Are there any ablation study results reflecting the performance gain brought by each part?
- The explanation of the formula content in Algorithm 1 in the main body is not clear enough.

[A] Temporal Coherent Test-Time Optimization for Robust Video Classification. ICLR23
[B] Video Test-Time Adaptation for Action Recognition. CVPR23

**Questions:**

- What does ω represent in Equation 3? Is it necessary to obtain it through additional training?
- How is the ground truth flow obtained in Figure 3? Is it supervised? This is not clearly described in the paper.
- In appendix A.3.1, the method is compared with the RAFT method (2020), which is out-of-date. Why not compare it with newer methods?